# Recombinant Rabies Virus Overexpressing OX40-Ligand Enhances Humoral Immune Responses by Increasing T Follicular Helper Cells and Germinal Center B Cells

**DOI:** 10.3390/vaccines8010144

**Published:** 2020-03-23

**Authors:** Yingying Li, Ling Zhao, Baokui Sui, Zhaochen Luo, Yachun Zhang, Yong Wang

**Affiliations:** 1Department of Human Anatomy, College of Basic Medicine, Dali University, Dali 671000, China; ayxwy76@163.com; 2State Key Laboratory of Agricultural Microbiology, Huazhong Agricultural University, Wuhan 430070, China; zling604@163.com; 3Department of Preventive Veterinary Medicine, College of Veterinary Medicine, Huazhong Agricultural University, Wuhan 430070, China; baokunsui1@163.com (B.S.); zcluo2012@163.com (Z.L.); zhangyachun20063024@126.com (Y.Z.)

**Keywords:** rabies vaccine, OX40-ligand, T follicular helper cells, germinal center B cells, plasma cells

## Abstract

Rabies, caused by the rabies virus (RABV), remains a serious threat to public health in most countries. Development of a single-dose and efficacious rabies vaccine is the most important method to restrict rabies virus transmission. Costimulatory factor OX40-ligand (OX40L) plays a crucial role in the T cell-dependent humoral immune responses through T-B cell interaction. In this work, a recombinant RABV overexpressing mouse OX40L (LBNSE-OX40L) was constructed, and its effects on immunogenicity were evaluated in a mouse model. LBNSE-OX40L-immunized mice generated a larger number of T follicular helper (Tfh) cells, germinal center (GC) B cells, and plasma cells (PCs) than the parent virus LBNSE-immunized mice. Furthermore, LBNSE-OX40L induced significantly higher levels of virus-neutralizing antibodies (VNA) as early as seven days post immunization (dpi), which lasted for eight weeks, resulting in better protection for mice than LBNSE (a live-attenuated rabies vaccine strain). Taken together, our data in this study suggest that OX40L can be a novel and potential adjuvant to improve the induction of protective antibody responses post RABV immunization by triggering T cell-dependent humoral immune responses, and that LBNSE-OX40L can be developed as an efficacious and nonpathogenic vaccine for animals.

## 1. Introduction

Rabies is a devastating inflammation targeting the central nervous system; it is a preventable disease, yet, it causes around 59,000 human deaths annually [1,2]. Rabies virus (RABV), the causative agent, is a non-segmented, single negative-stranded RNA virus that belongs to the *Lyssavirus* genus of the Rhabdoviridae family. More than 99% of human rabies is transmitted by dog bites or licks [3,4]; therefore, dog rabies control can possibly lead to a decline in human rabies cases. More than 70% of vaccination coverage of the canine population could eliminate rabies in humans [5]. However, animal inactivated vaccines with multiple-dose vaccination programs are not cost-effective, which hinders their extensive implementation in most countries [6]. Live-attenuated recombinant RABVs (rRABVs) can achieve protective immunity merely after a single dose; therefore, they are less costly and have potential to be developed as safe and cost-effective vaccines to control animal rabies [7,8]. Additionally, the rRABV expressing a cytokine or a chemokine has been reported to improve the induction of virus-neutralizing antibodies (VNA) by enhancing the immunogenicity [9,10,11]. Therefore, developing affordable live-attenuated rRABV, expressing an immunoregulatory factor, would be a feasible strategic approach to protect animals from rabies.

OX40-ligand (OX40L), a type II transmembrane protein, was found on activated T and B cells, and it had a high secretion level in myeloid antigen-presenting cells (APCs), including dendritic cells (DCs), macrophages, B cells [12,13], mast cells, and vascular endothelial cells [14]. Commonly, OX40L is efficient at augmenting the pool of antigen-specific CD4^+^T cells and subsequently up-regulating naïve and memory CD4^+^T cells in this pool to secrete multiple T follicular helper (Tfh) cell-associated molecules, which further effectively induced Tfh cell generation [15]. Additionally, OX40L signals with its receptor (OX40) played an important role in the T cell-dependent humoral immunity through the interaction between OX40-expressing activated T cells and OX40L-expressing activated B cells [16]. Previous studies showed that the interaction between OX40/OX40L and the inducible costimulatory molecule (ICOS)/inducible costimulatory ligand (ICOSL) was necessary for inducing Tfh cells and germinal center (GC) B cells, and for maintaining GC reactions to promote plasma cell (PC) generation and virus-specific antibody responses during vaccinia virus (VACV) immunization [17], suggesting that OX40L might be a potential adjuvant for vaccine development.

OX40L, used as an adjuvant in DNA vaccine, has been reported to be an effective strategy to induce humoral responses against pathogenic virus infection [18]. In this study, a rRABV expressing murine OX40L was constructed to evaluate its immunogenic properties and stimulatory effect on the humoral immunity by studying the T cell-dependent B cell immune response in a mouse model. Our results indicated that the rRABV expressing OX40L could promote protective antibody responses against RABV infection by increasing Tfh cells, GC B cells, and PCs.

## 2. Materials and Methods

### 2.1. Cells, Viruses, Antibodies, and Animals

The cell line BSR cells, a cloned cell line derived from BHK-21 cells, were cultured in Dulbecco’s modified Eagle’s medium (DMEM) (Gibco, Grand Island, NY, USA), containing 10% fetal bovine serum (FBS) (Gibco, Grand Island, NY, USA) and antibiotics (100 units/mL Penicillin and 100 µg/mL Streptomycin) (Beyotime, Wuhan, China). The cell line mouse neuroblastoma (NA) cells were cultivated in Roswell Park Memorial Institute (RPMI)-1640 medium (Gibco, Grand Island, NY, USA) containing 10% FBS and antibiotics (100 units/mL Penicillin and 100 µg/mL Streptomycin). The rRABV strain LBNSE was derived from SAD L16 (generated from the attenuated SAD-B19 vaccine strain) by removing the pseudogene and introducing *BsiW*I and *Nhe*I sites between the G and L genes, with mutation at amino acid sites 194 and 333 in the glycoprotein [19]. The rabies challenge virus strain 24 (CVS-24) was propagated in 5-day-old sucking mouse brains. Fluorescein isothiocyanate (FITC)-conjugated antibodies against the RABV N protein were purchased from Fujirebio (Malvern, PA, USA). These antibodies were used for detection of RABV-specific antibody isotypes, including horseradish peroxidase (HRP)-conjugated goat anti-mouse immunoglobulin G (IgG), IgG1, IgG2a, IgG2b, IgG3, and immunoglobulin M (IgM) (Boster, Wuhan, China). The following antibodies used for flow cytometric analysis, including FITC-CD4 (clone RM4–5), APC-CXCR5 (clone 2G8), PE-PD1 (clone J43), PE-Cy7-B220 (clone RA3-6B2), PE-CD95/Fas (clone Jo2), Alexa Fluor647-GL7 (clone GL7), and APC-CD138 (clone 281-2), were purchased from BD Biosciences (San Jose, CA). The 5-day-old suckling Institute of Cancer Research (ICR) mice, and 6-week-old female ICR mice and BALB/c mice were purchased from the Center for Disease Control and Prevention of Hubei Province, Wuhan, China. Ethical procedures for animal experiments were followed in accordance with guidelines approved by the Scientific Ethics Committee of Huazhong Agricultural University (protocol number HZAUMO-2015-029).

### 2.2. Construction and Rescue of rRABV Expressing Murine OX40L

The LBNSE-OX40L cDNA clone was constructed as previously described [19]. In brief, the murine OX40L cDNA was amplified from total RNA extraction from RABV-infected mouse brain tissues via RT-PCR. Two primers were designed for OX40L PCR production (forward primer: 5′-TTG CGT ACG ATG GAA GGG GAA GGG GTT C-3′ and reverse primer: 5′-CTA GCT AGC TCA CAG TGG TAC TTG GTT C-3′ with *BsiW*I and *Nhe*I sites underlined). The pLBNSE vector and OX40L PCR production were digested with *BsiW*I and *Nhe*I. Then, OX40L gene was cloned between G and L genes of pLBNSE to replace the deleted non-coding pseudogene, resulting in the generation of pLBNSE-OX40L. LBNSE-OX40L was rescued in BSR cells. These BSR cells were transfected with the infectious clone of pLBNSE-OX40L and four helper plasmids containing the N, P, G, and L genes of LBNSE by using SuperFect (Qiagen, Valencia, CA, USA) according to the instructions [19]. The rescued LBNSE-OX40L was confirmed using FITC-conjugated RABV N protein-specific antibodies.

### 2.3. Virus Titration

Direct fluorescent antibody test was used to determine RABV titers in BSR cells [19]. Briefly, BSR cells were incubated with a 10-fold serial dilution of rRABVs in 96-well plates at 37 °C; for 48 h. These 96-well plates were then washed 3 times with 300 µL phosphate-buffered saline (PBS) and fixed with 80% ice-cold acetone. The cells in the plates were then incubated with FITC-conjugated antibodies against RABV N for 45 min. Antigen-positive foci of infected cells were detected via an Olympus IX51 fluorescence microscope (Olympus, Tokyo, Japan). The RABV titers were presented as focus-forming units per milliliter (FFU/mL). Each virus dilution was added into four duplicated wells.

### 2.4. Cell Viability Assay

The BSR cells were incubated with LBNSE, LBNSE-OX40L, or DMEM (as mock) respectively, at multiplicity of infection (MOI) = 1 in 96-well microplates at 37 °C; for 48 h. Subsequently, the infected cell supernatants were collected for cell viability assay using cell proliferation assay kits (Promega, Madison, WI, USA) following the instructions. The optical density (OD) at 490 nm was obtained by a Spectra MAX 190 microplate reader (Molecular Devices, Sunnyvale, CA, USA).

### 2.5. OX40L Concentration Determination through ELISA

BSR cells (a clone of BHK-21 cells) were pre-seeded into 48-well microplates to reach 90%–95% confluency and inoculated with LBNSE or LBNSE-OX40L at a MOI = 1, 0.1, 0.01, or 0.001 at 37 °C; for 24 h. Supernatants were collected to quantify OX40L with commercially mouse OX40L ELISA kits (RayBiotech, Atlanta, GA, USA), according to the manufacturer’s instructions. OD value was obtained at 450 nm using a Spectra MAX 190 microplate reader.

### 2.6. Mouse Immunization and Challenge Test

A total of 30 female 6-week-old ICR mice were divided into three groups equally. The 100 μL volume containing 10^6^ FFU of LBNSE or LBNSE-OX40L, or an equal volume of DMEM (as mock) was vaccinated to the hind legs of mice through an intramuscular (im) route. All the immunized-mice from three groups were intracerebrally (ic) challenged with 30 µL of 50 lethal dose 50 (LD_50_) CVS-24 at 28 dpi. The clinical symptoms were carefully monitored and recorded twice daily after the challenge.

### 2.7. Pathogenicity Studies

Female 6-week-old ICR mice were injected via ic route with 30 μL of 10^7^ FFU of LBNSE, LBNSE-OX40L, or an equal volume of DMEM (as mock), respectively. The 5-day-old ICR mice were injected through ic route with 10 μL of 100 FFU above-mentioned rRABVs, respectively. Body weight and survival rates were carefully observed twice daily for 21 days.

### 2.8. Detection of Virus-Neutralizing Antibody (VNA)

The detection of VNA titers was carried out through the fluorescent antibody virus neutralization (FAVN) assay [20]. At the indicated time points, blood samples were collected, and then sera were isolated. The 50 μL of threefold serial dilution of serum samples or standard serum and 100 μL of cell culture medium were added into 96-well plates. Each well was coated with a 50 μL CVS-11 containing around 100 FFU at 37 °C; for 1 h. The cell suspensions containing 2 × 104 BSR cells were added into each well, and incubated at 34 °C; for 3 days. Then, the plates were fixed with 80% ice-cold acetone at −20 °C; for 15 min. These plates were washed three times with 300 μL PBS. FITC-conjugated anti-RABV N antibodies were added into each well and were incubated at 37 °C; for 45 min. Thereafter, positive signals were identified using an Olympus IX51 fluorescence microscope. Fluorescence values of serum samples were compared with the titer of a standard serum, and VNA titers were calculated, and presented as international units per milliliter (IU/mL). All blood samples were determined in quadruplicate.

### 2.9. RABV-Specific Antibody Isotype Test

The sera from immunized mice were isolated to analyze the RABV-specific antibody isotypes by ELISA. The 96-well ELISA plates were coated with 100 µL of 0.5 µg/well RABV G protein in coating solution (5mM Na_2_CO_3_, pH 9.6) at 4 °C; for 12 h. Then, the plates were washed three times with 300 µL PBS-Tween and blocked with 5% low-fat milk in PBS-Tween at 37 °C; for 2 h. Serum samples were diluted up to 1:30 in PBS-Tween. The dilution samples were added into each well and incubated at 37 °C; for 2 h. Post incubation, the plates were washed three times with 300 µL PBS-Tween. The 100 µL of goat anti-mouse HRP-conjugated IgG (1:1000), IgG1 (1:1500), IgG2a (1:1500), IgG2b (1:2000), IgG3 (1:1500), or IgM (1:2000) was added into the corresponding wells and incubated at 37 °C; for 45 min. The plates added with 100 µL of tetra-methyl-benzidine (TMB) substrate (Boster, Wuhan, China) were placed at 37 °C; for 30 min in the dark. Afterwards, 50 µL of 2M H_2_SO_4_ was added into each well to stop the reaction. The OD value at 450 nm was observed under a Spectra MAX 190 microplate reader.

### 2.10. Flow Cytometric Assay

Three BALB/c mice from each group were immunized through im route with 100 μL of 10^6^ FFU of LBNSE, LBNSE-OX40L, or an equal volume of DMEM (as mock). The inguinal lymph node (LN) cells and bone marrow (BM) cells were harvested at 7 and 14 dpi. Thereafter, cells were resuspended in 200 µL of 0.2% bovine serum albumin (BSA) and labeled with fluorescence-conjugated antibodies, and then placed at 4 °C; for 30 min. Afterwards, cells were washed two times with 2 mL PBS, and resuspended into 300 µl cell suspensions (10^5^ cells/sample) for analyses. Data were collected via a CytoFLEX flow cytometer (Beckman Coulter, Brea, CA, USA), and analyzed through CytExpert software (Beckman Coulter, Brea, CA, USA).

### 2.11. Statistical Analysis

All statistical analysis was carried out by GraphPad Prism 8 (GraphPad Software, lnc., San Diego, CA, USA). Survival rates were calculated using the Kaplan–Meier method [21]. Differences between survival curves were analyzed with the log–rank test. A two-tailed unpaired t-test was conducted to reveal statistical significant difference of the other data. Data were representative of two independent experiments. The statistical significant differences were presented at the levels of *, *p* < 0.05; **, *p* < 0.01; ***, *p* < 0.001.

## 3. Results

### 3.1. Characterization of rRABV Expressing OX40L

To evaluate the role of OX40L as an adjuvant in the RABV-induced immune responses, the murine OX40L cDNA was cloned into LBNSE vector. This rRABV was rescued as described previously [19], which was designed as LBNSE-OX40L (Figure 1A). The rRABV encoding the murine OX40L gene was stable for at least ten consecutive passages in BSR cells, which was confirmed by sequencing. The BSR cells (Figure 1B) and NA cells (Figure 1C) were treated at a MOI = 0.01 to develop the multiple step growth curves. The results obtained from the growth curves showed that no significant difference in the rRABVs titers was found at indicated time points on both cell types, suggesting that the extra OX40L insertion did not change viral replication in vitro. To determine whether OX40L could be successfully expressed by LBNSE-OX40L, we detected the concentration of OX40L in the infected cell supernatants by ELISA kits. ELISA results indicated that BSR cells infected with LBNSE-OX40L expressed OX40L in a dose-dependent manner, whereas those infected with LBNSE did not (Figure 1D). The results of cell viability assays suggested that the cell growth and viability of LBNSE-OX40L-infected BSR cells were not significantly different from those of the LBNSE-infected cells, indicating that the OX40L had no detrimental effect on the infected cells (Figure 1E).

### 3.2. Pathogenicity of rRABV Expressing OX40L

To determine whether overexpression of OX40L might cause any adverse effects on animals, three groups (10 female six-week-old ICR mice per group) were ic injected, respectively, with 10^7^ FFU of LBNSE, LBNSE-OX40L, or DMEM (as mock). Their body weights were monitored daily for 14 days. No rabies-related clinical symptoms were observed in any group. The LBNSE-OX40L-injected mice exhibited a slight increase in body weight compared with the LBNSE-injected mice, although this difference was not statistically significant (Figure 2A). To investigate the effect of LBNSE-OX40L on viral pathogenicity in an immunocompromised mice model, three litters of 5-day-old sucking mice ic received a total of 100 FFU of rRABVs mentioned above, respectively. Then, their survival rates were recorded. The survival rate of the mice injected with LBNSE-OX40L was slightly increased, compared to that of the mice injected with LBNSE (Figure 2B). Taken together, these data suggested that OX40L overexpression did not cause any undesirable effects in vivo.

### 3.3. Generation of Tfh Cells in Mice Immunized with LBNSE-OX40L

OX40L signals were reported to induce Tfh cells in the immune responses to VACV [17]. Based on it, the generation of Tfh cells was investigated after rRABVs immunization in this study. Three groups of BALB/c mice received 10^6^ FFU of LBNSE, LBNSE-OX40L, or DMEM (as mock) through im route, respectively. At 7 and 14 dpi, inguinal LN cell suspensions were collected and detected by flow cytometry. The population and quantity of lymphocytes in 10^5^ individual inguinal LN cells were shown in Figure 3A. The representative flow result of CD4^+^T cells (Figure 3B) and Tfh cells (Figure 3C) were shown [22]. We observed that Tfh cells in the LBNSE-OX40L-vaccinated mice were significantly more than those in LBNSE-vaccinated mice at 7 and 14 dpi (Figure 3D). Taken together, our results indicated that the overexpression of OX40L promoted the generation of Tfh cells post RABV immunization.

### 3.4. Generation of GC B Cells in Mice Immunized with LBNSE-OX40L

Since Tfh cells crucially contributed to the generation of GC B cells, the role of overexpression of OX40L in GC B cell induction was further investigated. At 7 and 14 dpi, draining LN cells were collected from the immunized-mice and stained to analyze GC B cells. Representative flow data on B220^+^B cells (Figure 4A) and GC B cells (Figure 4B) were shown [23]. The GC B cells were significantly accumulated in the mice receiving LBNSE-OX40L at 7 and 14 dpi (Figure 4C). These data indicated that OX40L displayed a stimulative property to induce GC B cells post RABV vaccination.

### 3.5. Generation of PCs in Mice Immunized with LBNSE-OX40L

PCs exist mainly in the BM, which are principally differentiated from GC B cells [24]. Antibodies induced by PCs play a critical role in protection from viral infection [25]. Considering this point, we further investigated the effect of OX40L on the induction of PCs. The 10^5^ BM cells were harvested from three groups of immunized mice at each indicated time points. The populations and number of investigated cells were shown in Figure 5A. The representative flow results of PC (B220^lo^CD138^+^) detection were shown in Figure 5B. Interestingly, significantly higher percentage of PCs in the immunized-mice BM cells was induced by LBNSE-OX40L than by LBNSE at 7 and 14 dpi (Figure 5C). These results illustrated that the overexpression of OX40L promoted the induction of PCs after RABV immunization.

### 3.6. Antibody Responses and Protection in Mice Immunized with LBNSE-OX40L

Since OX40L overexpression significantly augmented the percentage of PCs, the role of OX40L in antibody production was further evaluated. Blood samples from different rRABVs immunized mice were collected weekly to measure VNA titers by FAVN. The LBNSE-OX40L-immunized mice induced significantly higher VNA levels (as early as 7 dpi, lasting for 8 weeks) than the LBNSE-immunized mice (Figure 6A). The LBNSE-OX40L-immunized group first exhibited a dramatic increase in VNA levels, with a maximal geometric mean of VNA titer (GMT) of 42.36 IU/mL observed at 28 dpi, and then followed by a decrease. However, the GMT of LBNSE-immunized group reached the peak of 19.54 IU/mL at 21 dpi, and then gradually reduced (Figure 6B). After 63 dpi, no significant difference in the VNA titers was detected between the two groups. In addition to VNA titers, optical density (OD) values of different specific antibody isotypes against RABV G in serum samples were determined (Figure 6C). Consistently, the LBNSE-OX40L-immunized mice induced significantly higher levels of RABV G-specific IgG, IgG1, and IgG2a at all tested time points, and induced significantly higher levels of IgG2b at 7, 21, 28, and 35 dpi, than the LBNSE-immunized mice. The significant difference was observed in the levels of IgG3 and IgM only at 35 and 7 dpi, respectively.

To further investigate the consistence of better protection with the abundant antibody production, we evaluated the role of LBNSE-OX40L in protection against virulent RABV challenge. At 28 dpi, all of the immunized mice were challenged via ic route with 50 LD_50_ of CVS-24, then their clinical symptoms were monitored for another 21 days (Figure 6D). All DMEM-injected mice (as mock) succumbed to rabies within 14 days. The challenge experiment results that the LBNSE-OX40L-immunized mice exhibited a higher survival rate (87%) than LBNSE-immunized mice (47%) were in line with the detection results of antibody titers. Taken together, OX40L overexpression could augment antibody production and provide potent protection post RABV immunization.

## 4. Discussion

Vaccines induce antibody responses usually in two main manners, namely, T cell-independent and T cell-dependent B cell immune responses. It has been previously reported that the low and short-lived VNA titers induced by T cell-independent immune responses reached a peak within seven dpi [26]. Recently, the interaction between CD4^+^T cells and specialized B cell populations has been reported to be crucial in maintaining a high VNA level for a relatively long time [11]. Previous studies of rRABVs expressing a cytokine such as fms-like tyrosine kinase 3 ligand (Flt3L) showed that the immune factor contributed to enhancing humoral immune responses by mainly recruiting and activating dendritic cells (DCs), resulting in the substantial VNA production [27]. However, the process of the VNA production is relatively slow and the maintenance of antibody response is relatively short-term. In this study, the rRABV expressing OX40L (LBNSE-OX40L) was constructed to further confirm the role of T cell-dependent B cell response in enhancing humoral immune responses to RABV. Interestingly, LBNSE-OX40L induced significantly higher levels of VNA as early as seven dpi, and lasted up to eight weeks, resulting in a better protection than its parent virus LBNSE.

OX40L is a multi-functional immune modulator contributing to adaptive immune responses, especially to T cell and B cell proliferation and differentiation. Therefore, the potential of OX40L as a functional adjuvant applied in many viral vectors has been confirmed [18,28,29,30]. OX40L could lead to the expansion of antigen-specific CD4^+^T cell populations by promoting initial CD4^+^T cell activation, clonal division, and survival [31,32]. In addition, OX40L signals with its receptor were confirmed to be essential for upregulating CD4^+^ helper T cell populations to express multiple Tfh cell-associated molecules, such as cysteine-X-cysteine (CXC) chemokine receptor type 5 (CXCR5), ICOS, interleukin-21 (IL-21), and macrophage activating factors [33]. The interaction between OX40L signals and these Tfh cell-associated molecules stimulated the CD4^+^T cells to differentiate into Tfh cells [15,17,34]. OX40-OX40L interaction facilitated the activated CD4^+^T cells to migrate and accumulate at the border between T zone and B cell follicles, further resulting in formation of Tfh cells [31]. These previous findings provided the evidence that OX40L contributed to Tfh cell development. In the present study, OX40L overexpression induced the significant increase in Tfh cell populations, which were main supporters to the GC reaction.

This study demonstrated that the LBNSE-OX40L-immunized mice generated a large number of GC B cells. This result might be directly associated with the improved Tfh cells. The Tfh cells provided necessary stimuli for the differentiation and proliferation of GC B cells, and they were responsible for affinity maturation of B cell clones in the GCs [35,36]. One previous study showed that OX40L was found on the activated B cells, whose interaction with its receptor resulted in a dramatic increase in B cell proliferation [16]. The increased B cells might act in concert with the assistance of augmented Tfh cells, thereafter, these B cells promoted the GCs development [37,38]. Additionally, mice lacking OX40-OX40L signaling displayed the significantly reduced accumulation of GC B cells and Tfh cells, suggesting that OX40L signals were indispensable for the generation of GC B cells [17]. Therefore, with the assistance of the amplified Tfh cells, exogenous OX40L could further facilitate the accumulation of GC B cells in the increased B cell pool.

In our study, the LBNSE-OX40L immunization was able to induce quickly a potent antibody response (high levels of VNA and RABV G-specific antibody isotypes) as early as seven dpi, which might be due to the increased number of PCs at seven dpi, or even earlier. A previous mouse model study reported that the accumulation of PC populations was significantly reduced in OX40-deficient mice [17], indicating that OX40-OX40L signals were particularly important for the development of PCs. Previous study of 2,4,6 trinitro-phenyl-keyhole limpet hemocyanin immunization indicated that OX40L interaction with its receptor was also crucial for promoting terminal differentiation of the activated B cells into highly Ig-secreting PCs during the process of T cell-dependent humoral immunity [16]. On the other hand, the induction of GC B cell populations by OX40-OX40L interaction [17] might be related to the OX40L function of PC development. This interaction drove augmented GC B cells towards the differentiation of abundant long-lived PCs [17,39], which further produced a large number of antibodies over time [40]. In addition, consistent with a previous report [6], the high value of RABV G-specific IgG2a and IgG2b subclasses sustained by OX40L expression was associated with prolonged protective antibody responses post RABV immunization. Hence, LBNSE-OX40L was able to sustain robust VNA levels for up to eight weeks.

For the safety concern of using live-attenuated rabies vaccine, we compared the viral replication on BSR cells and NA cells through multiple-step growth curves and evaluated pathogenicity through body-weight changes in adult mice and survival rates in suckling mice. Some previous studies reported that functional CD8^+^T cell responses induced by OX40-OX40L signals were correlated directly with robust reduction of viral replication and attenuation of vaccinia virus (VACV) infection [41] and foot and mouth disease virus (FMDV) infection [42]. In our study, the multiple-step growth curves showed no statistically significant difference in the rRABV titers between LBNSE-OX40L and its parent virus on both cell types. Pathogenicity studies showed that LBNSE-OX40L-injected mice exhibited a slight improvement on body-weight loss in adult mice and survival rates in suckling mice compared to the LBNSE-injected mice. This non-significant attenuation might be explained by the report that the restriction on viral replication and virulence by OX40L signals was virus-specific [43]. Additionally, the parent rRABV LBNSE was derived from the SAD-B19 with mutation at amino acid sites 194 and 333 in the G protein [19,44], resulting in the nonpathogenicity of rabies vaccine for both im and oral immunizations [10,45]. Taken together, OX40L overexpression did not cause any undesirable side effects in vivo and LBNSE-OX40L could be a live-attenuated and avirulent rabies vaccine.

## 5. Conclusions

In conclusion, the rRABV expressing OX40L designed in this study exhibited an ability to induce robust humoral immune responses by increasing Tfh cells, GC B cells, and PCs. This LBNSE-OX40L has potential to be developed as an efficacious and nonpathogenic vaccine for animals.

## Figures and Tables

**Figure 1 vaccines-08-00144-f001:**
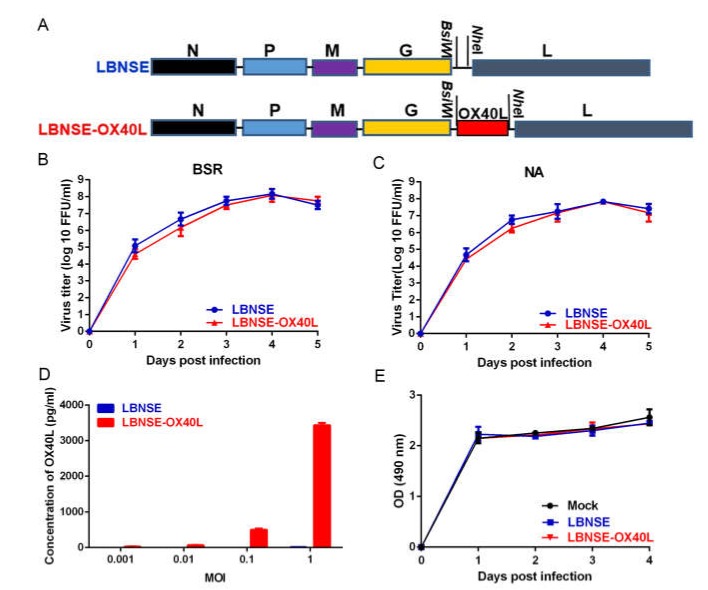
Characterization of recombinant rabies viruses (rRABV) expressing OX40-ligand (OX40L) in vitro. (**A**) Construction diagram of LBNSE and LBNSE-OX40L. The vector pLBNSE was derived from the pseudogene-deleted SAD-B19 strain. The *BsiW*I and *Nhe*I sites were introduced between G and L genes. N, P, M, G, and L represented RABV nucleoprotein, phosphoprotein, matrix, glycoprotein, and polymerase genes, respectively. Murine OX40L was inserted into this LBNSE genome in place of the deleted pseudogene. Supernatants of the cells infected with different rRABVs at a MOI = 0.01 were collected at 1, 2, 3, 4, and 5 dpi to determine the titers of LBNSE and LBNSE-OX40L. Based on it, multiple-step growth curves of LBNSE and LBNSE-OX40L, respectively, infecting BSR cells (**B**) and NA cells (**C**) were developed. (**D**) The expression level of mouse OX40L in the infected cell supernatants was detected by ELISA kits. (**E**) Detection of cell viability of rRABV-infected BSR cells. Error bars represented the standard deviation (SD, *n* = 3).

**Figure 2 vaccines-08-00144-f002:**
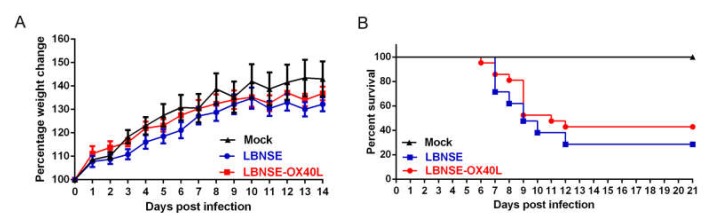
Pathogenicity of rRABV expressing OX40L. (**A**) Body weight changes of female six-week-old ICR mice (*n* = 10) infected with 10^7^ focus-forming units (FFU) of LBNSE, LBNSE-OX40L, or Dulbecco’s modified Eagle’s medium (DMEM) (as mock) through intracerebrally (ic) route. (**B**) Survival rates of three litters of five-day-old sucking mice (*n* = 21) receiving 100 FFU of above-mentioned rRABVs via ic route. Error bars represented the standard error (SE).

**Figure 3 vaccines-08-00144-f003:**
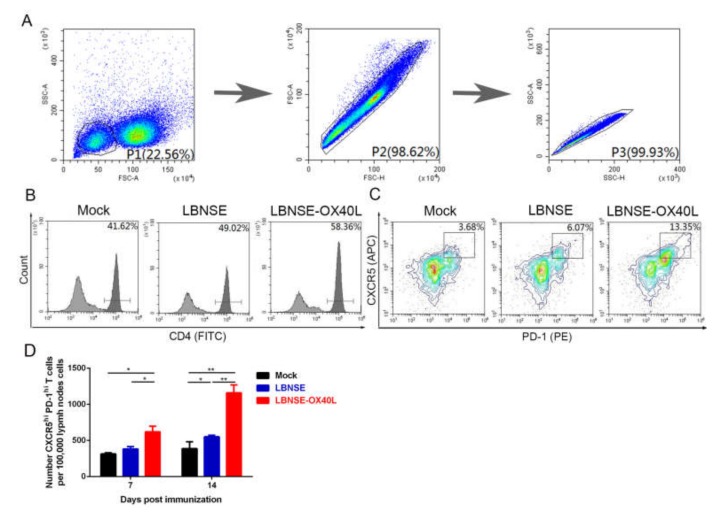
Generation of T follicular helper (Tfh) cells in mice immunized with LBNSE-OX40L. Moreover, 10^5^ single inguinal lymph node (LN) cells from the mice immunized with 10^6^ FFU of LBNSE, LBNSE-OX40L, or DMEM (as mock) were stained by antibodies for analyzing the Tfh cells (CD4^+^CXCR5^hi^PD-1^hi^) by flow cytometry. Cell populations of interest were visualized by pseudocolor plots (**A**) or histogram plots (**B**) or contour plots (**C**) showing their forward-scatter (FSC) and side-scatter (SSC) signals in relation to the size and granularity/complexity of the cells, respectively. The gating strategies and representative contour plots of lymphocyte populations in LN cells (A), CD4^+^T cell populations in lymphocyte (B), and Tfh cell populations in CD4^+^T cells (C) were shown. (**D**) At 7 and 14 dpi, the total number of Tfh cells per 10^5^ draining LN cells was determined. Error bars represented the SE (*n* = 3). The following notations were used to indicate significant differences between groups: *, *p* < 0.05; **, *p* < 0.01.

**Figure 4 vaccines-08-00144-f004:**
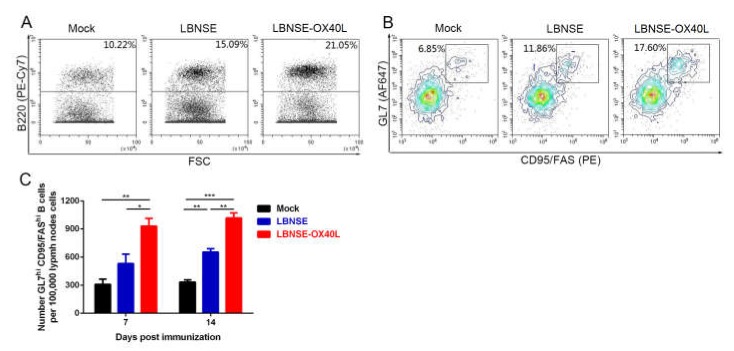
Generation of germinal center (GC) B cells in mice immunized with LBNSE-OX40L. The induction of GC B cells (B220^+^CD95^+^GL7^+^) was analyzed by flow cytometry, after the immunized-mice inguinal LN cells were labeled by antibodies. Cell populations of interest were visualized on dot plots (**A**) or contour plots (**B**) showing their FSC and SSC signals in relation to the size and granularity/complexity of the cells, respectively. The representative flow cytometric strategies of the B220^+^B cell populations in lymphocytes (**A**) and GC B cell populations in B220^+^B cells (**B**) were shown. (**C**) At 7 and 14 dpi, the total number of GC B cells per 10^5^ draining LN cells from mice immunized with rRABVs was determined. Error bars represented the SE (*n* = 3). The following notations were used to indicate significant differences between groups: *, *p* < 0.05; **, *p* < 0.01; ***, *p* < 0.001.

**Figure 5 vaccines-08-00144-f005:**
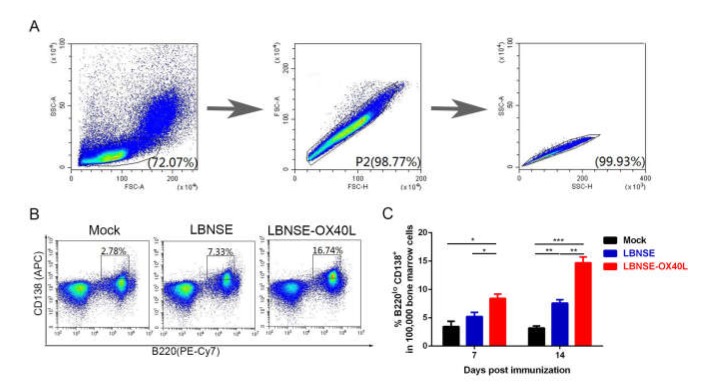
Generation of plasma cells (PCs) in mice immunized with LBNSE-OX40L. Moreover, 10^5^ single bone marrow (BM) cells from the immunized mice were collected and stained with PE-Cy7-B220 and APC-CD138 for analyzing the induction of PCs by flow cytometry. Cell populations of interest were visualized by pseudocolor plots (**A** and **B**) showing their FSC and SSC signals in relation to the size and granularity/complexity of the cells, respectively. The representative gating strategies and contour plots of lymphocyte populations in BM cells (A), and PC (B220^lo^CD138^+^) populations in BM lymphocyte (B) were shown. (**C**) At 7 and 14 dpi, the percentage of PCs within 10^5^ BM cells was analyzed. Error bars represented the SE (*n* = 3). The following notations were used to indicate significant differences between groups: *, *p* < 0.05; **, *p* < 0.01; ***, *p* < 0.001.

**Figure 6 vaccines-08-00144-f006:**
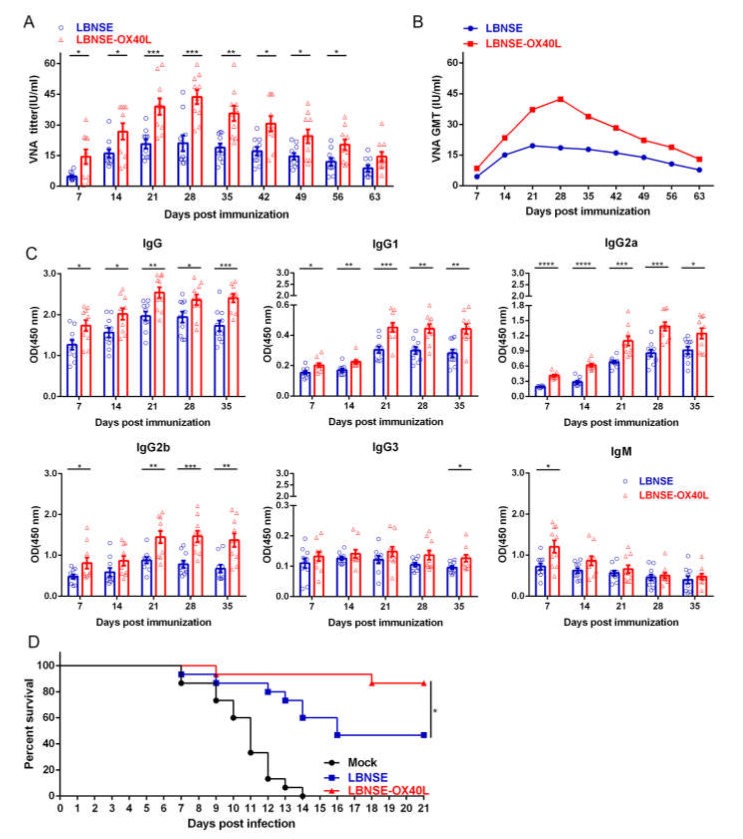
Antibody responses and protection in mice immunized with LBNSE-OX40L. ICR mice (*n* = 10) were immunized via intramuscular (im) route with 100 μL volume containing 10^6^ FFU of LBNSE or LBNSE-OX40L, or an equal volume of DMEM (as mock). Sera from vaccinated mice were collected weekly for nine weeks, and the virus-neutralizing antibodies (VNA) titers (**A**) and GMT (**B**) of sera were determined by using fluorescent antibody virus neutralization (FAVN). (**C**) The optical density (OD) values of total RABV G-specific immunoglobulin G (IgG), IgG1, IgG2a, IgG2b, IgG3, and immunoglobulin M (IgM) from immunized-mice sera were determined by ELISA. At 28 dpi, all the immunized mice (*n* = 15) were challenged with 50 LD_50_ of CVS-24 via ic route, and then they were carefully monitored daily for 21 days. The survival diagram was constructed according to the number of survivors in each group (**D**). Error bars represented the SE. The following notations were used to indicate significant differences between groups: *, *p* < 0.05; **, *p* < 0.01; ***, *p* < 0.001.

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
