# Peer review of "Recombinant Rabies Virus Overexpressing OX40-Ligand Enhances Humoral Immune Responses by Increasing T Follicular Helper Cells and Germinal Center B Cells"

_vaccines, 2020, doi:10.3390/vaccines8010144_

Round 1

Reviewer 1 Report

The manuscript "Recombinant rabies virus overexpressing OX40-ligand enhances humoral immune responses by increasing T follicular helper cells and germinal center B cells" reports the development of a recombinant rabies virus that expresses the immune modulating protein OX40 on infection. The authors find that when compared to the parent virus, the modified virus shares many properties (growth rate and attenuation in adult mice) but also shows increased stimulation of germinal center formation that is reflected in increased levels of virus neutralising antibodies and increased levels of virus-specific IgG subtypes. The manuscript is generally well written and the data supports the conclusions. Only minor corrections are suggested below:

Line 18. At some point define LBNSE.

Figure 1D. State in the figure legend or text  the significance of measuring IL-7.

Figure 6. Correct text line 292 (ethic 6 to Figure 6).

Line 356. Define VACV.

Line 371. LBNSE-OX40L has potential to be developed...

Author Response

Dear reviewer,

Sincerely,

Yingying Li, Ph.D

Reviewer 2 Report

This is an interesting paper about a recombinant RABV overexpressing mouseOX40L (LBNSE-OX40L) that was constructed and investigated for its effects on immunogenicity in a mouse model. LBNSE-OX40L-immunized mice generated larger number of T follicular helper (Tfh) cells, germinal center (GC) B cells and plasma cells (PCs) than the parent virus LBNSE-immunized mice. Furthermore, LBNSE-OX40L induced significantly higher levels of virus-neutralizing antibodies (VNA) as early as 7 days post immunization (dpi), which lasted for 8 weeks, resulting in better protection for mice than LBNSE.

Minor revisions are included in the text.

Author Response

(The authors gave the same response as above.)

Reviewer 3 Report

Review Report

ID Vaccines -755330

Title: „Recombinant Rabies Virus Overexpressing OX40-ligand Enhances Humoral Immune Responses by Increasing T Follicular Helper Cells and Germinal Center B Cells„

Author‘s: Yingying Li, Ling Zhao, Baokui Sui, Zhaochen Luo, Yachun Zhang, Yong Wang

Version: 1 Date: 17/3/2020

Reviewer number: 1

A brief summary

The manuscript „„Recombinant Rabies Virus Overexpressing OX40-ligand Enhances Humoral Immune Responses by Increasing T Follicular Helper Cells and Germinal Center B Cells„ Author‘s: Yingying Li, Ling Zhao, Baokui Sui, Zhaochen Luo, Yachun Zhang, Yong Wang

 provides fairly specific biotechnology-type information on the using a new type of adjuvant that potentially influences the production of rabies-specific antibodies and promotes a humoral immune response by using live attenuated vaccines for rabies vaccination. The objective (one main sentence) of the Article is not clearly identified in the “Introduction” part. However, at the end of the chapter we can find the “possible” tasks of the research article: to evaluate the immunogenic properties of the type II transmembrane protein (OX40L) and its stimulatory effect on humoral immunity by studying the T-cell immune response using a mouse model in vivo.

Broad comments

The research article may be interesting enough for a narrow range of scientific readers who are directly involved in biotechnology research on rabies vaccines (but not necessarily) and qualitative evaluation of vaccine immune response. The Article is written in a relative appropriate way, but can be better structured - it contain a lot of specific virology and molecular biology research that is very “correctly” presented in Materials and Methods, but difficult to "trace" in the Results. So, when the reader read the Results, he/she needs to analyze and evaluate this data all the time to properly understand what it means from a scientific point of view and why it is important for evaluating specific immune response.The Introduction prepared well with possibly focus on the objective of the study, however, prior to vaccination information (say L39), the introduction is not related to the intended purpose of the article. I would recommend expanding (suppose one-two extended sentences for each immune response factor) paragraph L46-L56 which provides very good general introductory information on the specific immune response and OX40L stimulating effect. The objective (s) of the scientific article must be precisely identified at the end of the introduction.The Materials and Methods are described very well and detailed enough, however, some locations require additional data (see specific comments). The Results are well presented, but their strategic scope is very large. In a classic scientific article, it is recommended that the results be presented in an analogous sequence of methods, but in this case it is very difficult to do so. Therefore, the reader may sometimes have questions about the added scientific value of the results and how those findings may support the conclusions of the presented research.

All the presented research results should be discussed in the article “Discussion” section – all!

The list of reference should be revised and unified according to the specific recommendation for authors.

Specific comments

L 29. Keywords. I recommend delete „humoral immune response“.

L 43. “Virus-neutralizing antibodies” insert “… VNA”.L 65. “BSR cells” additional specification is required (clone/line, T7/5 – the same?)

L 68. “Mouse neuroblastoma (NA)“additional specification is required (clone/line,…)

L 69. „(penicillin and streptomycin)“ additional specification is required (concentration of working/stock sol. or qualitative indicators of primary dilution).

L 101. After “…Olympus IX51” identification (producer, country…).

L 165. Ref. identification after “the Kaplan-Meier method…”.

Author Response

(The authors gave the same response as above.)
